# Roadkills as a Method to Monitor Raccoon Dog Populations

**DOI:** 10.3390/ani11113147

**Published:** 2021-11-04

**Authors:** Linas Balčiauskas, Jos Stratford, Laima Balčiauskienė, Andrius Kučas

**Affiliations:** Nature Research Centre, Akademijos 2, 08412 Vilnius, Lithuania; birding.jos@gmail.com (J.S.); laima.balciauskiene@gamtc.lt (L.B.); andrius.kucas@gamtc.lt (A.K.)

**Keywords:** *Nyctereutes procyonoides*, roadkill-based survey, hunting bag, density, Lithuania

## Abstract

**Simple Summary:**

The raccoon dog (*Nyctereutes procyonoides*) is a highly invasive species, therefore knowledge regarding the population size, trends and spatial distribution is important for species assessment and for the planning of control measures. In Lithuania, however, as such surveys would require the concerted action of many hunting clubs, raccoon dog surveys have not been carried out since 1997. In this study, we investigated whether roadkill data for the species could be used as a means for monitoring the population. For the period in which surveys were conducted, we found a strong correlation between the numbers surveyed and the numbers of animals hunted and between those hunted and the number of roadkills. Therefore, we consider that roadkill counts may be used as a proxy for a survey at the countrywide scale. Practical implementation of the method is proposed.

**Abstract:**

The raccoon dog (*Nyctereutes procyonoides*) is one of the most frequently killed species on Lithuanian roads. As an invasive species, up-to-date knowledge of population size, trends and spatial distribution is critically important both for species assessment and for the planning of control measures. In Lithuania, however, raccoon dog surveys have not been carried out since 1997. We investigated, therefore, whether roadkill counts on predefined routes could be used as a proxy for a survey. Our dataset includes survey numbers for the period 1956–1997, hunting bag sizes for 1965–2020 (including the spatial distribution of the hunting bag in 2018–2020) and roadkill data relating to 1551 individuals between 2002–2020. At the most local scale, that of the hunting areas of hunting clubs, correlations between the numbers of hunted and roadkilled individuals were negative and insignificant or absent. At the country scale, however, we found significant correlation both between the numbers surveyed and hunted in 1965–1997 (r = 0.88), and between those hunted and the number of roadkills in 2002–2020 (r = 0.56–0.69). Therefore, we consider that roadkill counts on predefined and stable routes may be used as a proxy for a survey at the country scale. Practical implementation of the method is proposed.

## 1. Introduction

For effective population management, reliable data on population size and/or density are essential. When such data are not available, trends in the relative abundances, expressed as indices, are very important for evaluating the effectiveness of management efforts. This is equally important for game, small game, large carnivores and invasive mammals. However, there is a cost element to the accuracy of data [1,2], with the costs of higher accuracy sometimes becoming unacceptable. New recently developed methods, such as drone surveys, camera counts and surveys based on DNA analysis, can be somewhat expensive, but equally so too can traditional groundwork by qualified persons [3,4].

Following on from this, the possibility of using hunting bag records as an indicator of the population size of selected species has been discussed for a long time (see [5,6]). This discussion on the suitability of the hunting bag (or hunting statistics) as a proxy for population status is still ongoing [7,8]. The general conclusion regarding the suitability of this method is positive, despite possible sources of bias, such as weather factors [9], habitat conditions [10], length of the hunting season [11], non-random choices made by hunters [8] and non-response of hunters regarding bag data [12]. Despite these factors, the overall consensus is that hunting statistics may be used as a complimentary method to surveys [13].

There are cases when biases within these statistics may be explainable. In Lithuania, the correlation between the raccoon dog (*Nyctereutes procyonoides*) population numbers and the hunting bag was negative in the period of 1984–1990. The first years of this period were known as the “golden years of the fur market”. Hunters earned good money supplying raccoon dog pelts. There was also additional hunting to supply a considerable black market in furs. The second reason for the increased intensity of hunting of raccoon dogs was the introduction and rapid expansion in the numbers of hunting terriers, these being able to chase raccoon dogs both above and underground very effectively. However, after 1990, at the beginning of country’s independence, the hunting industry declined and switched its focus primarily to ungulates; therefore, the raccoon dog hunting intensity significantly decreased. Later on, the fur market shrunk considerably and the hunting intensity of the species returned to the pre-1984 period (E. Tijušas, unpubl.). Another method to estimate the population numbers or densities of vertebrate species is through the use of roadkill counts as a proxy for these parameters [14,15,16]. It should be borne in mind however that considerable biases may exist in the data due to the varying duration and frequencies of road counts in different studies, some being done daily or weekly, others on a monthly or seasonal basis [17,18,19,20]. The methods of roadkill registration can also skew the quality and reliability of results, this frequently depending on whether data is obtained through rigorous scientific survey and/or police registrations [21,22], or through crowdsourcing or citizen science records [23,24]. Various environmental factors, such as land use, proximity of forest, etc. [22] can also impose bias on the data, as can changes in road management, traffic intensity and measures to reduce animal-vehicle collisions, such as fencing. However, even allowing for all of these possible biases, roadkill analysis is accepted as a valuable method for monitoring wildlife populations [25].

As a highly successful invasive mammal species [26], we focused our study on the raccoon dog. During the second half of the 20th century and the beginning of 21st century, this species spread from west Russia to western Europe, reaching Spain and the Netherlands [27]. However, contrary to that stated by [28], the species is not established in Great Britain (https://www.gbif.org/species/2434552, accessed on 12 September 2021). To date, the raccoon dog has already become established in 27 countries in Europe and is still expanding its range. As a result, it is listed as an EU species of concern [28]. Though formerly managed as a valuable source of pelts, the current goal of management is its eradication, therefore knowledge of trends in the population is required to determine the effectiveness of the eradication program. In Lithuania, despite the need for such information, raccoon dog surveys have not been conducted since 1997, thus there is a lack of information on the population numbers or trends in abundance. Though rarely included in the roadkill database of the Lithuanian Police Traffic Supervision Service, it is the case that raccoon dogs are the most numerous roadkill species as registered by professional biologists in Lithuania [21].

As shown by Saeki and Macdonald [29], raccoon dog roadkills, after adjustment for traffic intensity, may be used as a proxy for population trend. To assess this, we analysed data of raccoon dog numbers according to the earlier surveys, to hunting records in the country (both at the district and hunting area levels) and to roadkills of the species. Using hunting records at the best spatial resolution, we tested whether a correlation existed with the roadkill distribution and, if so, how strong it was. Based on Schwartz et al. [25], we aimed to discover whether the distribution and numbers of raccoon dog roadkills, registered on predefined routes and then overlaid with the hunting statistics, could be used to monitor population numbers and trends.

## 2. Materials and Methods

### 2.1. Study Site

Racoon dog roadkill counts and hunting records covered the entire territory of Lithuania, a northern European country situated at the Baltic Sea coast (Figure 1a) and located in the mixed forest zone. The country has a continental climate with an average yearly temperature of 6.1 °C and rainfall between 490–850 mm. The human population density is 45.3 inhabitants/km^2^. The main habitats in the country are agricultural land and forests, these being 52.26% and 33.2%, respectively (http://www.nzt.lt/go.php, accessed on 15 June 2020). The road network in the country amounts to 21,238 km (http://lakd.lrv.lt/en/sector-activities/road-network, accessed on 12 September 2021).

### 2.2. Data Collection

As the first and main source of data for the raccoon dog roadkill statistics, we used professional registrations, i.e., collected by biologists, covering the period 2007–2020 on main roads (length 1750.7 km, annual average daily traffic (AADT) 9413 vehicles/day in 2017), national roads (4927.7 km, AADT 2261 vehicles/day in 2017) and regional roads (14,559.2 km, AADT 390 vehicles/day in 2017). The second source, the database of the Lithuanian Police Traffic Supervision Service (2002–2020), did not contribute much to the overall roadkill number (Figure 2), as the raccoon dog is too small to cause human casualties and thus such roadkills were generally only registered where required to obtain payments relating to car insurance, this mostly being only in the last few years of this period. As it was shown by Balčiauskas et al. [21], the Lithuania police data underestimates small and medium-sized animal roadkill numbers, with the database primarily targeted to accidents involving ungulates. Using both sources, data was gathered on 1551 roadkilled raccoon dog individuals in the period 2002–2020 (Figure 1b). The roadkill index (individuals killed per km of route driven) for the species was calculated using Nature Research Centre data only.

The third data source was that of the game survey, this being the national surveys for the period 1956–1997 and national hunting bag sizes for 1965–2020. Collected from the Ministry of Environment, these are the so-called “official” statistics (Appendix A). Finally, a fourth data source were hunting bag records at the level of hunting areas, this giving spatial distribution in the hunting seasons 2018/2019–2020/2021 (courtesy of the Environmental Protection Department under the Ministry of Environment). In 1956–1997, ca 300,000 raccoon dog individuals were surveyed, while ca 155,000 raccoon dogs were hunted in the period 1965–2020 using both firearms and traps. The locations of the hunting areas were available for a total of 1852, 1756 and 1303 raccoon dogs, hunted in the 2018/2019, 2019/2020 and 2020/2021 seasons, respectively. We did not use random selection of hunting data according to [8]. The reporting of the raccoon dog hunting bag is related to obtaining the quotas for limited game, thus not every club did this every year (if they were not requesting the limits, the bag was not represented), but we did not treat such situations as non-response as according to Aubry and Guillemain [12].

### 2.3. Statistical Analyses

A General Linear Model (GLM) test was conducted to assess the influence of road type (categorical character) on the raccoon dog roadkill index (dependent parameter), while temporal data variability was controlled by the year and month of the survey, these being continuous predictors. F statistics were used to assess the significance of the model, while eta-squared was used for the influence of the single factor.

Before using GLM and ANOVA, we checked the normality of the raccoon dog roadkill using Kolmogorov–Smirnov D and excluded empty routes without registrations. Based on the mixed conformity to normal distribution (Appendix A), we chose one-way ANOVA as a robust test against the normality assumption, tolerating violations with only a small effect on the Type I error rate (https://statistics.laerd.com/statistical-guides/one-way-anova-statistical-guide-3.php, accessed on 14 July 2021).

We used Spearman correlations to find (i) if surveyed numbers were related to the hunting bag of the same year, (ii) if hunting bag was related to the roadkill, and (iii) if roadkill index was related to the sampling effort. Again, we used the assumption that the correlation coefficient is not dependent on the normality of the data (https://stats.stackexchange.com/questions/3730/pearsons-or-spearmans-correlation-with-non-normal-data, accessed on 25 July 2021).

Yearly and monthly proportions of non-null counts were compared using the G-test online calculator (https://elem.com/~btilly/effective-ab-testing/g-test-calculator.html, accessed on 2 August 2021).

## 3. Results

The normality tests of the yearly and monthly racoon dog roadkill distributions showed mixed results, as the roadkills in some years and all months of the spring–autumn period did not conform to normal distribution (Appendix A). The years 2002–2006 were represented by single cases only, these registered by the Lithuanian Police Traffic Supervision Service (Figure 2).

The highest proportion of routes with roadkilled raccoon dogs were characteristic to the years of small to moderate survey effort (Appendix A). The G-test with G = 125.87 showed that there were differences between the years with no less than *p* < 0.0001 confidence. The years 2018 and 2019, both with G = 18.74, can be characterised by the smallest proportion of the positive survey with *p* < 0.002.

Comparing the proportion of the routes with roadkilled raccoon dogs by month (Appendix A), we found the sample to be very heterogenic (G = 296.83, *p* < 0.0001), with the minimum proportion (7.3–11.3%) of positive routes being in the winter months and the maximum proportion (43.3–44.1%) of positive routes being in August and September (G = 47.10, *p* < 0.0001).

### 3.1. Temporal and Road Type-Based Trends of Raccoon Dog Roadkills

Although highly significant (F_5,3229_ = 25.0, *p* < 0.0001), the GLM model explained only 3.6% of the raccoon dog index variance. Individually, all tested factors were significant at *p* < 0.001, with the strongest influence being study month (F_1,3229_ = 84.6, eta-squared = 0.026) and the weakest being study year (F_1,3229_ = 15.2, eta-squared = 0.005) and road type (F_1,3229_ = 6.4, eta-squared = 0.006).

By year, the highest raccoon dog roadkill indices were recorded in 2012 (0.009 ± 0.003 ind./km/day), significantly exceeding those in 2018 and 2019 (Tukey honestly significant difference (HSD), *p* < 0.02), and in 2013 (0.011 ± 0.001 ind./km/day), exceeding those in 2015–2020 (HSD, *p* < 0.01–0.001). Low roadkill indices in 2002–2006 should not be compared with the other years, as both registration methods were not used in these years. Decreased roadkill indices in 2017–2019 (Figure 3) were significantly lower than those in 2008, 2009, 2013 and 2014 (with *p* < 0.05 or higher). By month, the highest raccoon dog roadkill indices were recorded in August (0.014 ± 0.001 ind./km/day) and September (0.012 ± 0.001 ind./km/day), significantly (*p* < 0.05–0.001) exceeding those recorded in other months (Figure 3).

Finally, roadkill indices on the main roads characterised by higher AADT (0.007 ± 0.0004 ind./km/day) exceeded those on the national roads (0.005 ± 0.0004 ind./km/day, Tukey HSD, *p* < 0.005), but did not exceed those on regional roads (0.008 ± 0.003 ind./km/day).

We also checked whether the yearly sampling effort was correlated with the average raccoon dog roadkill index (Appendix A). While the number of the driven routes was very moderately correlated with the roadkill (r = 0.37), the total length of the route per year and the average length of the route was not correlated (r = 0.10 and r = 0.13 respectively, none of the correlations significant). The raccoon dog roadkill index was moderately correlated with monthly sampling (Appendix A), namely the number of driven routes (r = 0.54) and the total length of the route (r = 0.52), but not significantly (*p* = 0.07 and *p* = 0.08, respectively). As a result, we cannot interpret these trends as promoting minimum sampling effort as the intended method.

### 3.2. Population Numbers, Hunting Bag and Roadkills of the Raccoon Dog at the Country Scale

At the country scale, the correlation between the population numbers and the hunting bag in the period 1965–1997 was positive (r = 0.88, *p* < 0.001). For the graphic representation, refer to Appendix A. However, there was a period in which this relationship was the opposite (1984–1990, r = –0.38). If this period is excluded, the dependence of the hunting bag on the population size is linear (r = 0.96, *p* < 0.001).

The presence of this dissonant period did not allow a true approximation of the population size from the hunting bag data alone, as the proportion of hunted animals may have significantly differed even if the number of hunted animals remained the same (Appendix A). We are not able to state whether the current situation is closer to the 1984–1990 period or to the years either side of this period. Thus, with the hunting bag being close to 2000 individuals, the proportion of the population that were hunted could be 15–20%, but could also be 40%, i.e., differing hunting intensities. The analysis of changes of hunting intensity, however, is beyond of the scope of the current study. Furthermore, no raccoon dog surveys were carried out after 1997.

The roadkill index, averaged for the country scale, was significantly correlated with the hunting bag in 2002–2020 (r = 0.50, *p* < 0.05), mainly based on the correlation of the roadkill index with the main roads (r = 0.56, *p* < 0.025). In the period 2007–2020, with increased roadkill count intensity, these correlations were higher (r = 0.54, *p* < 0.05 and r = 0.62, *p* < 0.025 respectively). In the years later than 2012, the correlations were similar for the main and national roads (r = 0.70 and r = 0.69, both *p* < 0.05), while the roadkill and hunting bag correlated (r = 0.64) for the regional roads, but not significantly (Figure 4). At the country scale, roadkill can be used as a basis for the prognosis of the next year hunting bag (Appendix A).

### 3.3. Raccoon Dog Hunting Bag and Roadkills at the Local Scale

At the level of hunting areas, the distribution of the number of hunted raccoon dogs and the densities of hunted animals in 2018–2021 was not even, with the highest numbers in the eastern part of the country (Figure 5). Comparing the numbers of the roadkilled raccoon dogs with the numbers (or densities) of hunted individuals in the hunting areas, no significant correlations were found (Pearson’s r = 0.06–0.10, all NS). Variation of the hunting bag and density of hunted individuals was similarly high in every year of the 2018–2020 period (Appendix A), and no correlations were found (r = −0.01–−0.02).

Dynamics of the raccoon dog roadkill index on the A14 main road by year and by month (Figure 6) in general corresponded to that at the country scale (see Figure 3); the highest indices were recorded in 2009 (0.005 ± 0.001 ind./km/day), 2013 (0.006 ± 0.001 ind./km/day) and 2020 (0.003 ± 0.0003 ind./km/day), as well as in August (0.007 ± 0.002 ind./km/day) and September (0.007 ± 0.001 ind./km/day). The unevenness of roadkill distribution was well expressed (Appendix A).

The proportion of null counts on the A14 main road was not significantly different across years (G = 10.0, NS), despite the totals varying widely at a national level—for example, 25.0, 37.5, 29.3 and 18.4% of countrywide routes recorded roadkilled raccoon dogs in the years 2009, 2013, 2015 and 2017, while no roadkills were registered in 2011, 2012 and 2019 (the latter due to COVID-19 pandemic travel restrictions). At the level of months, raccoon dog roadkills were not registered on the A14 in December and January, while the proportion of roadkilled raccoon dogs differed significantly on this route in the other months (G = 36.80, *p* < 0.001), with the maximum proportion being in August and September (44.2 and 50.0% respectively), these significantly exceeding February to July (G = 17.49, *p* < 0.005).

## 4. Discussion

A variety of direct and indirect methods may be used to monitor the presence of large and medium-sized mammals and to evaluate their density indices, these including direct observations, trapping, hunting with known effort, transect counts, camera traps, the use of scent stations, field sign counts and hair traps. Some of these may be conducted by hunters or professional biologists, while others by citizen scientists [30,31,32,33,34,35,36,37,38].

For raccoon dog monitoring, the usage of track counts or evaluation by excrement surveys is of limited value in the northern part of the middle latitudes where the animals hibernate [26]. For Lithuania, we considered the possibility of direct counts by hunters and citizen scientists, but such counts would require intensive concerted action by many hunting clubs in order to obtain non-biased and robust number estimation, as observations should be done at the same time to avoid data overlapping. Our experience of using hunters for large carnivore monitoring showed that coverage of the entire country’s territory was not possible even for key species such as the wolf and lynx [39,40].

We propose the monitoring of raccoon dog populations and trends through the changes seen in a certain index—in our case, this being the roadkill intensity of the species per known monitoring effort (individuals/km/day), based on fixed routes and the comparison of the index values over time. The roadkill index, obtained with measurable effort (length of the route), may be further developed into the number of killed animals per day and extrapolated to a yearly number roadkilled. A similar approximation of population based on roadkill records, traffic intensity and habitat features was presented by Tatewaki and Koike [16]. However, as in our study, using a fixed route for the survey removes all biasing factors [41] and we expect it to be suitable for large territories without already known difficulties [32]. The main presumption of the method, i.e., that the roadkill index is correlated with population size, has been confirmed for many other mammal species [14,42,43,44]. Specifically, it has been shown that roadkill registration is an acceptable method for assessing the long term trends of another omnivorous invasive mammal, that of the racoon (*Procyon lotor*), over wide geographical areas [45].

We propose that the method is easily adaptable to be used by citizen scientists. The only qualifications required are the ability to recognize raccoon dogs (or to be ready to take pictures of every roadkilled individual found) and to have the possibility to drive the required routes, or at least parts of, in August and September, i.e., during the period with the maximum of roadkills. With the help of citizen scientists—for example, by those persons using some routes for regular commuting—assessment of roadkills may be extended on the regional road network. Such an approach would be most promising as it includes local inhabitants and their knowledge [24,46]. Various social groups have already been used for monitoring medium-sized [47,48,49] and large carnivores. In the case of carnivores, large spatial scales require significant resources and effort, therefore citizen science can be a helpful tool [50,51]. It certainly may be used for roadkill monitoring [52].

We accept that roadkill places are not stable and depend on population changes of the animals, this mainly influenced by local decreases due to hunting and roadkills, and local increases, as pointed out by Zimmermann Teixeira et al. [53]. As shown by the raccoon dog roadkill index, roadkilled numbers in Lithuania may significantly exceed the hunting bag [14].

## 5. Conclusions and Practical Implementation

We propose that, depending on the nature of the roadkill distribution and the resulting so called hotspots, roadkill monitoring effort should embrace systematic (continued in a routine way) and random (registering routes selected to cover long distances with a diversity of natural factors and habitats) aspects. With the fixed long routes, also including the vicinities of cities, the method will cover different risks of medium-sized species being roadkilled on the roads with varying traffic intensities [20,54,55]. The proposed routes for raccoon dog monitoring cover both various habitats and the full range of traffic intensities, these shown as being the two main factors, other than population size, responsible for roadkill volume [43]. Therefore, we conclude that the proposed method will reflect actual changes to the population size when comparing the roadkill index values over time. To date, with the exception of part of the A14 main road, which was surveyed on a weekly basis (with some exceptions) in the period 2009–2021, raccoon dog roadkill counts have not been conducted on a regular basis [21].

The four proposed routes for raccoon dog population monitoring via roadkills, each designated by a different colour, are presented in Figure 7. They cover a total of ca 3700 km of roads (1080 km, 950 km, 890 km and 770 km for the blue, yellow, red and green routes, respectively). The highest consideration of coverage is given to the main roads (1600 km) and national roads (1900 km), while regional roads are underrepresented (ca 200 km) due to low AADT and lower speed. However, the roadkill index obtained for raccoon dogs on the regional roads was not statistically different to that on the main roads. Therefore, for monitoring purposes, extension of route length and including an additional 5000–6000 km of the regional roads is not worthwhile. Some of the regional roads are still gravel roads, characterized by low speed of traffic and few roadkills [21]. With minor changes, the route was used for the 2008, 2019 and 2020 roadkill counts (the changes depended on roadworks); we worked on the assumption that major roadworks minimised the possibility of roadkills, thus these routes were changed to the next available roads of the same category.

Monitoring is foreseen for the second half of August and second half of September (for best representation, a third repetition may be carried out in the second half of October). One team of two persons (driver and observer) can cover these four routes in eight days, or two such teams in four days. The route should be started early in the morning before many of the roadkilled animals are cleared from the road.

Results (trends in the roadkill index) are intended to be used by managers as an independent measure to evaluate the success of management targeted at raccoon dog extermination or the diminishing of population numbers. A decreasing roadkill index would show the effectiveness of such measures, while a stable or increasing index should give cause to re-think the management strategy. For species with a longer history of roadkill data collection in Lithuania, the correlation of roadkill numbers and known population numbers is much higher (r = 0.94) than the correlation for the raccoon dog [56]. We are sure, therefore, that accumulating more data will lessen errors in the model.

For the future, after applying the proposed roadkill monitoring in 2021 and several subsequent years, we plan to re-test the regression between the roadkill index and the number of hunted animals to discover whether this relationship could be approximated as non-linear (regarding the 2012–2020 roadkill data, no better fit had been found for exponential, power or quadratic functions so far). Then, using the average and CI of the roadkill index, the numbers of hunted animals and the relationship between hunted animals and population size (see Appendix A), we would present a simple toolbox in the form of a table on how current index values may be interpreted in terms of raccoon dog population size and possible prognosis for the following year.

## Figures and Tables

**Figure 1 animals-11-03147-f001:**
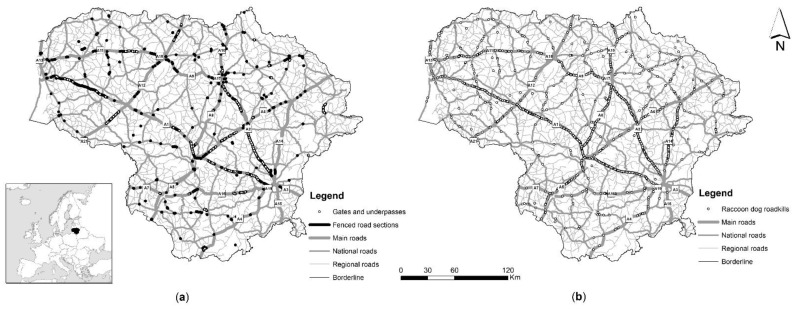
Road network of Lithuania (**a**) and registered raccoon dog roadkills in 2002–2020 (**b**).

**Figure 2 animals-11-03147-f002:**
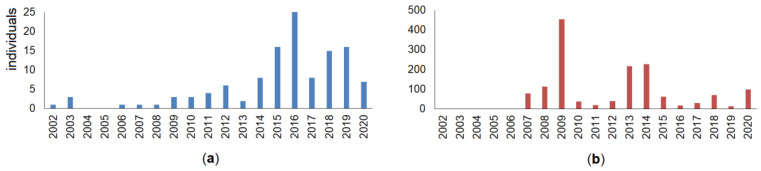
Raccoon dog roadkill data (individuals per year), registered by the Lithuanian Police Traffic Supervision Service in 2002–2020 (**a**) and the Nature Research Centre in 2007–2020 (**b**), datasets presenting different results both in terms of numbers and in temporal distribution of roadkills.

**Figure 3 animals-11-03147-f003:**
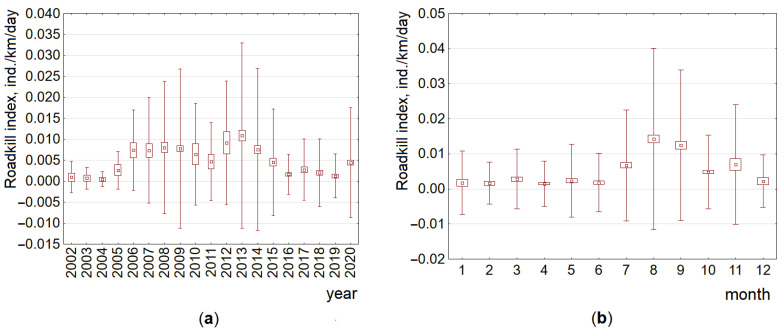
Raccoon dog roadkill trends by year (**a**) and by month (**b**) at the country scale, 2002–2020. Data from the Lithuanian Police Traffic Supervision Service in 2002–2020 and the Nature Research Centre in 2007–2020 are pooled and presented as averages, SE (box) and SD (whiskers).

**Figure 4 animals-11-03147-f004:**
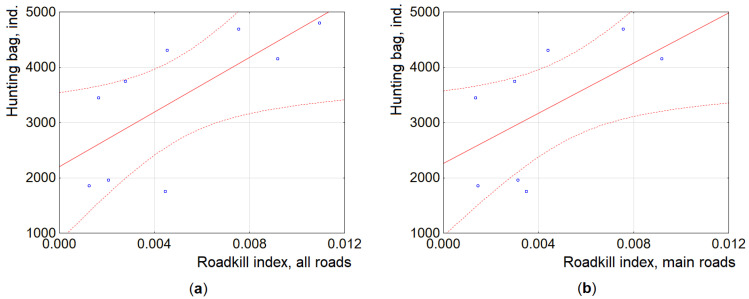
Correlation of roadkill index and raccoon dog hunting bag at the country scale, 2012–2020, presenting averages for all types of roads (**a**) and main roads (**b**). Dashed lines show 95% CI.

**Figure 5 animals-11-03147-f005:**
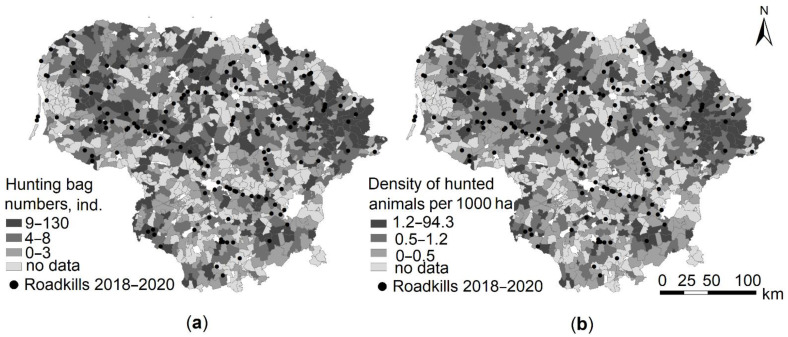
Spatial distribution of raccoon dog hunting bag numbers (**a**) and density of hunted animals (**b**) compared to roadkills at the local level in 2018–2020. White colour denotes territories of towns and cities, excluded from the hunting areas.

**Figure 6 animals-11-03147-f006:**
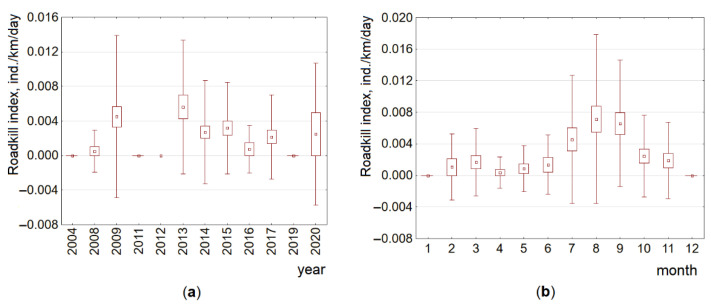
Raccoon dog roadkill trends by year (**a**) and by month (**b**) at the local scale on the A14 main road, 2004–2020. Data presented as averages, SE (box) and SD (whiskers).

**Figure 7 animals-11-03147-f007:**
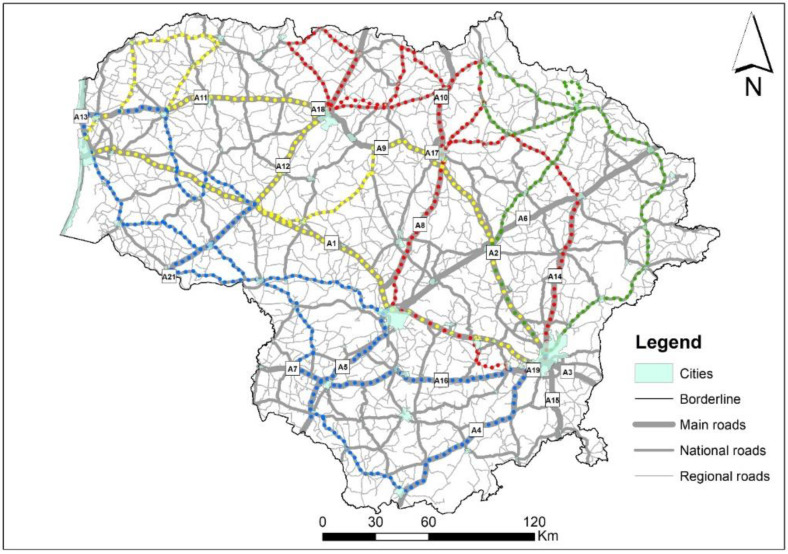
Proposed routes for raccoon dog roadkill monitoring in Lithuania, each designated by a different colour.

## Data Availability

Due to ongoing investigation, data of this study are available from the corresponding author upon personal request.

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
