# Peer review of "Roadkills as a Method to Monitor Raccoon Dog Populations"

_animals, 2021, doi:10.3390/ani11113147_

Round 1
Reviewer 1 Report
Detailed comments are included in the text. Generally, it should be clearly stated for which periods were actually correlated the data sets. Otherwise it is difficult for the reader to find out what was the size of analysed datasets. The text would benefit from a consultation with a native speaker.

Author Response
Rev#1 comments and answers
Comment: Lines 21-23, i.e. three temporarily independent data sets. How they can be correlated?
It is necessary to clearly declare which period was actually used for the analysis
Answer: Actual period is now shown in the Abstract.
Comment: caption of figure 2, it is obvious that both datasets give completely different results both in numbers and in temporal distribution
Answer: Caption corrected as advised: Figure 2. Raccoon dog roadkill data (individuals per year), registered by the Lithuanian Police Traffic Supervision Service (a) in 2002–2020 and the Nature Research Centre (b) in 2007–2020, datasets presenting different results both in terms of numbers and in temporal distribution of roadkills.
Comment: caption of figure 3, according to TSS or NRC?
Answer: We added an explanation to the caption: Data from the Lithuanian Police Traffic Supervision Service in 2002–2020 and the Nature Research Centre in 2007–2020 are pooled and presented as averages, SE (box) and SD (whiskers).
Comment: Line 236 “positive routes”????
Answer: Lines 236, 240 – by “positive routes” we meant routes where roadkilled raccoon dogs were found. The text has been changed, thank you for this comment, as it really was not clear.
Comment: Line 271, i.e. during the period with a maximum of roadkills - this should be clearly stated
Answer: Suggestion accepted, text added at Line 271
Comment: Line 290, why do not select a representative section of the road for every hunting district and use it as a permament transect for registration of roadkills?
Answer: We discussed such possibility in 2008, but there were several issues against such a decision.
- First of all, the average area of the hunting district is ca 6600 ha, and only a some of them have a road other than very a local road passing through it. Having irregular shapes of the hunting districts (see Figure 5), one club would have only a few kilometers of road in its area, therefore roadkill would be a random and occasional event.
- Secondly, hunters were not interested in registering raccoon dog roadkills, and still are not. Therefore, we could not guarantee that all hunting clubs would do the job. They have agreed to carry out snow track monitoring, and have been doing it every winter for the last three years. It was not the hunters idea, but a task asked by the Ministry of the Environment.
- Taking eight days for driving the permanent route is much easier and shorter in time in comparison to organizing efforts to ensure hunter participation.
- Finally, professionally done monitoring of the roadkill is less biased than citizen science (or monitoring done by hunters).
- Therefore, for the moment, we would like to stay with the proposed method, and will continue to work towards including citizen scientists – especially those commuting longer routes regularly – and we see such an approach as more promising.
Comment: Line 299, what is the meaning of those colours? the given lenght of various road categories is different from that belonging to blue, red and green. this is confusing
Answer: Apologies, the colours are just to show the routes clearly, as without colours it is difficult to distinguish the routes, especially at intersections. We have added explanation to the text at Lines 297–300, and in the caption of Figure 7.
Reviewer 2 Report
Several previous papers (Rolley and Lehman, 1992, Wild. Soc. Bull.; Case, 1978, Wildl. Soc. Bull.) have indicated highway speeds and use influence the number of road-kills in raccoons (Procyon lotor) a similar-sized species. The authors should report on this phenomenon in their analyses.
Are the densities of the target species known? It seems imperative that this is known to develop a comparative method of assessing a population through road-kill due to the above mentioned velocity and use influeincing road-kills.
Author Response
Rev#2 comments and answers
Comment: Several previous papers (Rolley and Lehman, 1992, Wild. Soc. Bull.; Case, 1978, Wildl. Soc. Bull.) have indicated highway speeds and use influence the number of road-kills in raccoons (Procyon lotor) a similar-sized species. The authors should report on this phenomenon in their analyses.
Answer: We inserted citation of the Rolley and Lehman, 1992, at Line 267.
Comment: Are the densities of the target species known? It seems imperative that this is known to develop a comparative method of assessing a population through road-kill due to the above mentioned velocity and use influeincing road-kills.
Answer: As we wrote in the Introduction, “In Lithuania, despite the need for such information, raccoon dog surveys have not been conducted since 1997, thus there is a lack of information on the population numbers or trends in abundance.”
We do not pretend to give densities of the raccoon dogs by monitoring of the roadkills. Our method will let us assess trends “through the changes seen in a certain index – in our case, this being the roadkill intensity of the species per known monitoring effort”. This index also might be “further developed into the number of killed animals per day and extrapolated to a yearly number roadkilled.”
Reviewer 3 Report
This article, “Roadkills as a method to monitor raccoon dog populations*” *by**
Linas Balčiauskas, Andrius Kučas, Laima Balčiauskienė and Jos Stratford. Is very interesting and much fun for me.I had never heard of raccoon dogs.I spent some time examining literature on raccoon dogs.
The paper is well written and had I had a Word version, I may have made some minor editorial comments.As it is, all comments are on sticky notes on the original manuscript.
*General comments:*
According to literature, raccoon dogs are considered invasive species.In North America, invasive species are usually considered for eradication even if they are species like Mountain Goats transplanted into places, they had never been.Such may not be the case in Lithuania.Hunting clubs are mentioned in the MS and “bags” like ducks in NA.It should be made very clear: are raccoon dogs valuable game animals that are managed for sustained harvest or are they pests to be eradicated? The goal of the census is different for each goal.If eradication is the goal the census is to determine the effectiveness of the eradication program.If raccoon dogs are considered valuable game animals, then the census is used to produce a sustainable harvest. You absolutely must make clear, for at least the North American readers.
When you mention hunting clubs, do hunters use firearms or do they trap?Make clear.
Out of curiosity, what do hunters do with collected raccoon dogs? You should make that clear for NA readers as well.
I would use the term “collected” rather than “registered”.Also, at least one of the figures is confusing or inadequately labeled (see sticky notes).
I think you made it clear that for the method to work, traffic, season, weather etc. must be fairly even among periods of census. You will never get a density but you may get some valuable data on population trends such as increasing or decreasing.Trends are what are mainly used in NA for managing big game animals.

Author Response
Rev#3 comments and answers
Comment: The paper is well written and had I had a Word version, I may have made some minor editorial comments. As it is, all comments are on sticky notes on the original manuscript.
Answer: Thank you, we have copied all your sticky notes from the pdf version and have answered these one by one below .
General comments:
Comment: According to literature, raccoon dogs are considered invasive species. In North America, invasive species are usually considered for eradication even if they are species like Mountain Goats transplanted into places, they had never been. Such may not be the case in Lithuania. Hunting clubs are mentioned in the MS and “bags” like ducks in NA. It should be made very clear: are raccoon dogs valuable game animals that are managed for sustained harvest or are they pests to be eradicated? The goal of the census is different for each goal. If eradication is the goal the census is to determine the effectiveness of the eradication program. If raccoon dogs are considered valuable game animals, then the census is used to produce a sustainable harvest. You absolutely must make clear, for at least the North American readers.
Answer: Information on the invasive status of the raccoon dog is presented in Lines 61–67. In the EU, this species is being eradicated by all possible means. We accept your comment that the goal of the population management was not clearly indicated, and therefore expanded the text in Line 68. “Though formerly managed as a valuable source of pelts, the current goal of management is its eradication, therefore knowledge of trends of the population is required to determine the effectiveness of the eradication program.”
Comment: When you mention hunting clubs, do hunters use firearms or do they trap?Make clear.
Answer: Both, in Lithuania raccoon dogs are hunted using firearms and trapped. We added an explanation to Line 119. However, this is not related to the roadkill registration.
Comment: Out of curiosity, what do hunters do with collected raccoon dogs? You should make that clear for NA readers as well
Answer: Hunters are informed only when large game (moose, deer, wild boar) are roadkilled and then they collect the carcasses. For raccoon dogs, hunters are not informed, so they do not collect carcasses. However, this is out of the scope of the paper – we are preparing a paper on the numbers of roadkilled animals where we can give the information you asked. If you mean hunted raccoon dogs, most of these go to sanitary pits, see the next comment.
Comment: I would use the term “collected” rather than “registered”. Also, at least one of the figures is confusing or inadequately labeled (see sticky notes).
Answer: Thank you, we have made changes to more figures than that one you mention. However, the term “registered” is correct – hunters do not collect roadkilled raccoon dogs from the road. Even hunted raccoon dogs are not usually skinned for the pelt, as there is almost no demand for it. The corpses are disposed of in special pits according to the rules of hunting (every club must have at least one such pit for the corpses of smaller animals and the intestines of the bigger ones).
Comment: I think you made it clear that for the method to work, traffic, season, weather etc. must be fairly even among periods of census. You will never get a density but you may get some valuable data on population trends such as increasing or decreasing. Trends are what are mainly used in NA for managing big game animals.
Answer: Thank you, we fully agree and in the revised text this is clearly stated. As there is no census data, roadkill index and approximation of the number of roadkilled animals indeed could add information necessary to plan management measures.
Comments from the text:
Comment: Line 33, some times abundance is difficult but trends (increases or decreases) can be critical to judging management efforts.
Answer: Thank you, we accept this comment and have added text after Line 34 “When such data are not available, trends of the relative abundances, expressed as indices, are very important to evaluate effectiveness of management efforts”
Comment: Line 69, Can you use "conducted" rather than "carried out"?
Answer: Sure, thank you for the proposed change
Comment: Line 70, or trends in abundance
Answer: Changed as suggested
Comment: Line 95, define professional
Answer: Here “professional” means “collected by biologists”, we have added this explanation
Comment: Figure 2, you need to lable the Y and X axis for clarity
Answer: Done for Y axis; we expect “year” on X axis should be clear without labeling
Comment: Line 118, clarify that "individuals" refers to raccoon dogs
Answer: Changed as suggested, in Lines 117–119.
Comment: Line 190, where are these "S figure"?
Answer: All supplementary materials were presented at the end of the manuscript text, after the References section. If accepted, supplementary Tables and Figures will be published in a separate file for download via the link.
Comment: Figure 4, Identify the roadkill indexes in each of these graphs. The one on the right indicates a national scale. What does the one on the left indicate?
Answer: Changes done, the left one is the general index for all types of roads, while the right one is for main roads. Caption and figure updated.
Comment: Line 234, You should use "among" and "between" only when comparing between two.
Answer: Thank you, this was changed to “across” in the revised text.
Comment: Line 256, define "this". I assume you refer to the lack of possibilities of using hunting information.
Answer: The beginning of the sentence changed to “We propose to monitor raccoon dog populations and trends through the changes seen in a certain index …”
Comment: Line 258, define stable routs.
Answer: we meant fixed, permanent – i.e. the same route used for registration. Word changed.
Comment: figure 7, What do colors represent?
Answer: We explained this as “The four proposed routes for raccoon dog population monitoring via roadkills, each designated by a different colour, are presented in Figure 7. They cover a total of ca 3700 km of roads (1080 km, 950 km, 890 km and 770 km for the blue, yellow, red and green routes respectively).” And “Figure 7. Proposed routes for raccoon dog roadkill monitoring in Lithuania, each designated by a different colour.”
The colour is just to see the configurations and positions of the routes.
Round 2
Reviewer 2 Report
The authors make a strong case for their methodology. However, I feel that without a known density from which to start, it will be difficult to assess trends. That is my belief. Other than that, the paper should be published.